# E5 Oncoprotein: A Key Player in Human Papillomavirus-Positive Head and Neck Cancer Pathogenesis and Therapy Resistance

**DOI:** 10.3390/v17040512

**Published:** 2025-04-01

**Authors:** Vanessa Emanuelle Pereira Santos, Bianca de França São Marcos, Pedro Henrique Bezerra Fontes, Micaela Evellin dos Santos Silva, Stephanie Loureiro Leão, Gabriel Rômulo Parente da Silva, Davi Emanuel Ribeiro, Marco Antonio Turiah Machado da Gama, Beatriz Eda de Oliveira Isídio, Ingrid Andrêssa de Moura, David Beltrán Lussón, Lígia Rosa Sales Leal, Aldo Venuti, Antonio Carlos de Freitas

**Affiliations:** 1Laboratory of Molecular Studies and Experimental Therapy, Department of Genetics, Federal University of Pernambuco, Av. Prof. Moraes Rego, 1235., 50670-901 Pernambuco, Brazil; vanessa.emanuelle@ufpe.br (V.E.P.S.); bianca.saomarcos@ufpe.br (B.d.F.S.M.); pedro.bezerrafontes@ufpe.br (P.H.B.F.); micaela.evellin@ufpe.br (M.E.d.S.S.); stephanie.lleao@ufpe.br (S.L.L.); gabriel.romulo@ufpe.br (G.R.P.d.S.); davi.ribeiro@ufpe.br (D.E.R.); marco.turiah@ufpe.br (M.A.T.M.d.G.); beatriz.eda@ufpe.br (B.E.d.O.I.); ingrid.andressa@ufpe.br (I.A.d.M.); david.beltran@ufpe.br (D.B.L.); 2HPV-Unit, UOSD Tumor Immunology and Immunotherapy IRCCS Regina Elena National Cancer Institute, 00144 Rome, Italy; aldo.venuti@ifo.it

**Keywords:** HNC, human papillomavirus, immune evasion, oncoprotein, tumor microenvironment

## Abstract

Head and neck cancer (HNC) is the sixth most prevalent type of cancer worldwide and is associated with low five-year survival rates. Alcoholism and smoking are the main risk factors associated with the development of head and neck cancer (HNC). However, Human Papillomavirus (HPV) infection has been reported as a significant risk factor, particularly for the oropharyngeal subset. In these cases, patients with HPV-positive HNC exhibit a better clinical prognosis; however, resistance to chemotherapy has been frequently reported. The carcinogenic activity of HPV is related to the viral oncoproteins E5, E6, and E7. E5 has been associated with immune evasion mechanisms and modulation of the tumor microenvironment, which appears to be linked to the virus’s resistance to chemotherapeutic treatments. Here, we review the potential of HPV E5 in targeted therapy for HNC and discuss relevant data regarding the activity of this oncoprotein in head and neck carcinogenesis.

## 1. Introduction

Head and neck cancer (HNC) is the sixth most common type of cancer worldwide and, despite advancements in targeted therapies, it has a five-year survival rate of less than 50% [1,2]. The use of conventional chemotherapy combined with immunotherapeutic drugs emerges as an alternative to minimize side effects in patients with HNC [2,3]. Alcohol and smoking are the main risk factors for the development of head and neck cancer (HNC). However, the increasing incidence of this cancer is significantly associated with Human Papillomavirus (HPV), particularly in oropharyngeal cancer [1,3,4]. This virus is the primary etiological agent of cervical cancer; however, in the United States, HNC is the most prevalent type of HPV-associated cancer [5].

HPV is one of the most prevalent viruses worldwide, infecting epithelial cells where its replication cycle occurs [6,7]. Generally, HPV-positive oropharyngeal subset HNC patients show better responses to chemotherapy compared to HPV-negative patients; however, a subset of HPV-positive patients does not respond to chemotherapy and experience cancer recurrence [1,8]. Therapeutic approaches against HPV-positive HNC are constantly being explored, including therapeutic vaccines of various modalities and agents that activate HPV-specific immune cells [4].

HPV infection alters immune cell infiltration in HNC, promoting a diverse and heterogeneous microenvironment [9]. The mediators of HPV’s resistance to cell death are the oncoproteins E5, E6, and E7 [10]. Studies related to HPV treatment focus especially on the E6 and E7 oncoproteins, including efforts aimed at therapeutic vaccination [11]. Here, we review data on the mechanisms of HPV E5 in mediating therapy resistance and highlight important considerations for targeted therapy in HPV-positive HNC cancers.

## 2. HPV and the Carcinogenic Process

HPV is the most common sexually transmitted viral infection worldwide, with most infections being transitory [12]. Chronic high-risk HPV infection is associated with 90% of cervical cancers, as well as a large proportion of anogenital cancers and head and neck cancers [13]. In addition, factors such as immune deficiency, immunosuppression, genetic predisposition, and environmental factors also contribute to mutations occurring in a prolonged infection [14].

When infecting cells, HPV can be found in episomal (extrachromosomal), integrated, or mixed forms, and the integration rate is commonly related to the level of DNA damage. The worse prognosis associated with HPV-positive head and neck cancers has been linked to the complete integration of the virus into the host genome, which is associated with reduced expression of tumor suppressors and limited infiltration of cytotoxic T cells, directly affecting the tumor microenvironment [15].

Through the integration of viral DNA into the host genome, malignant transformations occur in cells. The E6 and E7 proteins, which are oncogenic and promote carcinogenesis, allow the virus to have the ability to remain in keratinocytes in vitro and are also related to chronic HPV infection. By penetrating cervical epithelial cells and expressing the E6 and E7 proteins, the regulatory protein p53 and the retinoblastoma protein (pRb) are inactivated, which are essential in controlling the cell cycle and preventing apoptosis [14,16].

### 2.1. Structure and Viral Oncoproteins

HPVs are double-stranded DNA viruses belonging to the Papillomaviridae family [17,18]. Currently, there are more than 200 types of HPV classified according to their oncogenic potential into low-risk HPVs, related to benign lesions, and high-risk HPVs, related to carcinogenesis [19,20]. Their genomic structure is divided into three regions: a late region (L) encoding the L1 and L2 proteins responsible for the viral capsid structure; an early region (E) where the proteins E1, E2, E3, E4, E5, E6, and E7 are encoding and perform functions related to viral survival and replication, and a long control region (LCR), where gene regulation occurs and the origin of replication is located [21].

The oncoproteins E6 and E7 are the main players in the carcinogenesis process. E6 interacts with the LxxLL motif of E6AP, leading to the recruitment and polyubiquitination of the tumor suppressor p53 [22]. This oncoprotein can induce cell proliferation, inflammation, angiogenesis, and metastasis through the activation of the NF-κB transcription factor signaling pathway and increased expression of COX-2 [23]. The E7 oncoprotein induces the degradation of the retinoblastoma protein through ubiquitination, leading to the release of the E2F factor, which drives the cell into the synthesis phase, thereby contributing to cell cycle dysregulation [24]. Moreover, E7 can hinder cell differentiation by interacting with the C-terminal phosphatase domain of PTPN14, causing its degradation via ubiquitination [25].

The E6 and E7 oncoproteins interact with various signaling pathways, such as the PI3K/AKT pathway, promoting a stem cell phenotype in esophageal squamous cell carcinoma, and the JAK/STAT pathway, which is overexpressed due to HPV action in various cancers [26,27]. In recent years, the E5 protein has been extensively studied due to its role in immune evasion, cell growth, and therapy resistance, making it another important mechanism by which HPV can modulate the tumor microenvironment [28,29].

### 2.2. E5 Oncoprotein Characteristics

The E5 oncoprotein is not present in all types of HPV; this gene is specifically found in Alphapapillomavirus and Deltapapillomavirus, with Alphapapillomavirus being the only human papillomaviruses that harbor E5 [30]. The E5 protein can be subdivided into four distinct groups: α, β, γ, and δ. The E5α subgroup includes the major high-risk HPV types associated with cervical and penile cancers, such as HPV16 and HPV18. In contrast, the E5γ subgroup comprises the principal low-risk HPV types, such as HPV6 and HPV11, which encode two variants of the E5 protein (E5A and E5B) [31,32].

The E5α protein of HPV16 is the most extensively studied variant, with a molecular weight of 9.4 kDa (Figure 1A). It contains three hydrophobic regions organized into transmembrane domains and is predominantly localized in the endoplasmic reticulum, Golgi apparatus, and cytoplasmic membrane [32,33]. These subcellular localizations suggest that E5α is involved in protein trafficking, likely due to its ability to form pores and interfere with endocytic vesicle transport [34].

HPV genes are expressed from bicistronic or polycistronic pre-mRNAs that undergo alternative splicing, a process tightly regulated by the viral life cycle and host cell differentiation [35,36]. Early-region viral proteins are translated from mature mRNA polyadenylated at the pAE site, whereas late-region proteins are produced via polyadenylation at the pAL site [37] (Figure 1B).

Among the early proteins, the E5 oncogene is present in all mature mRNAs, including those encoding late proteins. Additionally, E5 is the fourth open reading frame (ORF) to be expressed, owing to its position at the 3′ end of the early-region genome [38]. However, the expression of HPV16 E5 varies depending on the mRNA isoform produced. This expression is initiated from the early promoter p97 and involves specific splicing events that generate transcripts containing the initial region of E6 and the E5 ORF (Figure 1) [39]. Notably, this mRNA is primarily produced during the early stages of the viral life cycle in HPV16-infected cells [40].

These findings highlight the crucial role of the E5 oncoprotein in HPV biology and its potential contribution to viral pathogenesis. Given that high-risk HPV types, especially HPV16, are strongly associated with head and neck squamous cell carcinomas (HNSCCs), understanding the role of E5 in these malignancies is crucial. In the next section, we will explore the epidemiology and general characteristics of head and neck cancer, emphasizing the impact of HPV infection and the potential contributions of viral oncoproteins, including E5, to its pathogenesis.

## 3. Head and Neck Cancer: Epidemiology and General Characteristics

Malignant tumors of the head and neck encompass a variety of neoplasms that arise in the regions of the upper aerodigestive tract, categorized according to anatomical location [41]. About 90% of these cancers are squamous cell carcinomas, which originate in the epithelial lining of the mucosal surfaces of the head and neck [42,43]. Due to the diversity of tissues present in these areas, the disease can originate in the larynx, nasal cavity, oral cavity, paranasal sinuses, pharynx, and salivary glands [44]. Together, these account for around 830,000 of cancer diagnoses worldwide and approximately 430,000 deaths per year, with variations depending on geographic region and demographic characteristics [45,46].

The World Health Organization (WHO) classification of head and neck tumors is the most widely used pathological classification system and is considered the gold standard for diagnosing these tumors. According to the WHO classification, tumors of the nasal cavity, paranasal sinuses, and skull base include sinonasal papillomas, mesenchymal tumors of the sinonasal tract, among others. Tumors of the nasopharynx include nasopharyngeal carcinoma, nasopharyngeal papillary adenocarcinoma, among others. In addition, there are other classifications, each with their respective subclassifications, such as tumors of the hypopharynx, larynx, trachea, and parapharyngeal space; tumors of the oral cavity and mobile tongue; tumors of the oropharynx (base of the tongue, tonsils, adenoids); salivary gland tumors; odontogenic and maxillofacial bone tumors; ear tumors; and paraganglionic tumors [47].

### Human Papillomavirus as a Risk Factor for Head and Neck Cancer

Despite the reduction in tobacco use in some countries, especially developed ones, the incidence of certain types of HNC has increased, attributed to human papillomavirus (HPV) infection [48,49]. HPV is responsible for 72% of HNSCC cases in developed countries, while in developing countries, the prevalence is 13% [45]. The incidence of the virus associated with this type of cancer is mainly observed in young individuals, extending the occurrence of cancer beyond older patients, who are commonly associated with other risk factors such as prolonged smoking [50]. Furthermore, the number of HPV-associated oropharyngeal cancer cases has exceeded the number of cervical cancer cases in developed countries, with the incidence varying regionally and between sexes, as oral HPV infection is more prevalent in men [51].

A systematic review and meta-analysis by Fonsêca et al. (2023) revealed a global HPV prevalence of 42% in oropharyngeal squamous cell carcinomas and 10% in oral squamous cell carcinomas, confirming infection trends in different cavities [52]. Additionally, the study noted that HPV16 was responsible for the majority of positive cases in both types of HNC, followed by HPV33, HPV58, and HPV18 [53]. The development of HNC is influenced by HPV through the infection of the virus in epithelial cells of the oral cavity and tonsils, which may undergo immortalization due to the influence of the viral oncoproteins expression [54].

There is heterogeneity in the expression of the HPV E5 oncogene among patients with HPV-positive tumors, which has significant implications for immune regulation and clinical outcomes. A study conducted in patients with head and neck cancer demonstrated that high E5 expression correlates with the downregulation of multiple major histocompatibility complex (MHC) class I genes, including HLA-B, HLA-C, and HLA-F, as well as decreased expression of the Antigen Peptide Transporter 2 (TAP2), which is crucial for antigen processing [29]. Furthermore, gene expression subtype analyses have classified tumors into two main profiles: E2/E4/E5-high and E6/E7-high, with the former group exhibiting lower expression of immunoproteasome subunits such as PSMB8 and PSMB9, potentially impairing antigen presentation and contributing to a more suppressed immune response [8]. This differentiation in viral expression patterns reflects the biological plasticity of HPV-positive tumors, where dominant E2/E4/E5 expression represents an alternative carcinogenic pathway independent of viral DNA integration into the host genome [55]. However, despite these insights, clinical studies investigating the role of E5 remain limited, with challenges including small patient cohorts, the complexity of HPV expression patterns, and the lack of standardized methods for detecting E5 expression in tumors. Collectively, these findings suggest that heterogeneity in E5 expression may impact tumor immune evasion and influence therapeutic response, underscoring the need for molecular subclassification to improve prognostic stratification and guide targeted therapeutic strategies. In the following sections, we will explore in greater depth the role of E5 in head and neck cancers, highlighting its mechanisms of action and potential clinical implications.

As described by Shigeishi (2023) [56], in HPV-positive oral cancers, mutations in the PIK3CA gene are frequently observed, while there are somatic mutations in ZNF750, FGFR3, DDX3X, CASZ1, PTEN, and CYLD, differentiating them from HPV-negative oral cancers. Furthermore, methylation and hypermethylation in genes such as MGMT, DAPK, CDKN2A, and others involved in Wnt/β-catenin or EGFR signaling pathways in oral cancer cells highlight the epigenetic regulation of HPV oncoproteins in these cancers. 

Although HPV is considered a risk factor for HNC, it has been associated with increased retention of B cells in the tumor microenvironment, which may be related to better prognosis in HPV-positive HNC patients [57]. According to the results obtained by Zhang and colleagues, the presence of HPV may stimulate the existence of B cells for surveillance and antibody production [57].

## 4. Activity of E5 in Head and Neck Cancer

### 4.1. Cellular Proliferation

The oncogenic activities of E6 and E7 are supported by E5, which also possesses its own oncogenic properties. Unlike E6 and E7, the E2/E4/E5 genes do not integrate into the host genome but are expressed on episomes in epithelial cells, and they can increase cell proliferation dependent on p53 and enhance cancer susceptibility [55]. E5 induces mitogenesis through various mechanisms, including the activation of MAPK via pathways dependent on and independent of the epithelial growth factor receptor (EGFR). The HPV E5 oncoprotein enhances EGFR signaling through direct and indirect mechanisms, promoting cell proliferation and cell cycle progression in cervical cancer (Figure 2) [38].

Activation of EGFR resulted in increased VEGF mediated by E5 which leads to increased activation of downstream pathways, including phosphoinositide 3-kinase (PI3K), AKT threonine kinase, and mechanistic target of rapamycin (mTOR) (Figure 2) [58]. Additionally, Sung Ho Um et al. observed in HPV-infected head and neck tumors that high E5 expression and low EGFR activity were associated with improved recurrence-free survival and overall survival [59]. Interestingly, the differential expression levels of HPV E5 in head and neck cancer (HNC) and cervical cancer (CC) suggest distinct oncogenic roles in these malignancies. In HPV-positive oropharyngeal cancer, E5 expression was found to be highly variable, with some tumors exhibiting significantly higher E5 levels than others. This variability was linked to increased EGFR expression, suggesting a potential role for E5 in modulating growth factor signaling and tumor progression [59]. Notably, tumors with high E5 expression demonstrated improved recurrence-free survival, whereas high EGFR expression correlated with worse clinical outcomes. This contrasts with findings in cervical cancer, where E5 has been implicated in enhancing viral persistence and immune evasion rather than directly influencing tumor aggressiveness. The distinct expression patterns of E5 in these cancers underscore the need for further research to elucidate its precise oncogenic mechanisms and potential as a biomarker for disease prognosis and treatment stratification.

Moreover, signaling mediated by Keratinocyte Growth Factor Receptor/Fibroblast Growth Factor Receptor 2b (KGFR/FGFR2b), an element related to the balanced maintenance of cellular differentiation and proliferation [60], was found to be suppressed due to the activity of the EGFR pathway which was increased by HPV18 E5 expression in cervical cancer [61]. HPV16 E5 has also been shown to induce epithelial–mesenchymal transition by negatively regulating FGFR2b along with the expression of a mesenchymal FGFR2c isoform [62]. This interference in signaling pathways further aids in the establishment of cellular immortalization and tumor progression (Figure 2).

### 4.2. Immune Evasion

PD-L1 is the primary ligand for its co-inhibitory receptor, PD-1, which is expressed on the membrane of activated T and B cells. PD-L1 is overexpressed in various cancers and plays a role in the negative regulation of the immune response by inhibiting the activation of CD4+ and CD8+ T cells [63]. The E5 protein can modulate PD-L1 levels and subsequently affect EGFR, which in turn promotes the expression of the Yes-associated protein (YAP). This upregulation of YAP enhances the expression of PD-L1, which is associated with T cell apoptosis. This leads to an increase in the death of CD8+ T lymphocytes responsible for the cytotoxicity of HPV-infected tumor cells [64,65,66]. Furthermore, the E5 oncoprotein has been identified as a mediator of resistance to the PD-L1 checkpoint blockade in head and neck cancers. The PD-L1 checkpoint negatively regulates MHC I expression, thereby interfering with antigen presentation and suppressing the immune response [29].

The HPV E5 gene plays a crucial role in evading immune surveillance during HPV infection due to its ability to negatively regulate MHC I molecules, specifically HLA-A and -B (Figure 2) [67,68,69]. Some mechanisms associated with this function include the retention of MHC I molecules in the Golgi apparatus due to the interaction between the HLA I heavy chain and HPV16 E5, which prevents the proper trafficking of these molecules in the Golgi apparatus and endoplasmic reticulum [70,71]. Additionally, E5 is capable of modulating H+ ATPase activity, preventing the acidification of endosomes and the Golgi apparatus, thereby compromising the processing capacity of these organelles [72,73].

E5 also interacts with the A4 endoplasmic reticulum protein, which regulates proliferative potential during differentiation [74]. CD1d, a glycoprotein that functions as a non-classical class I antigen-presenting molecule to Natural Killer T cells [75], found in antigen-presenting cells (APCs) and epithelial cells [76], was negatively regulated by the expression of HPV6 and HPV16 E5 in both in vitro and in vivo experiments. This regulation was attributed to the retention of CD1d in the endoplasmic reticulum due to the interaction of E5 with the chaperone protein calnexin [77].

E5 oncoprotein was able to downregulate the production of Interferon K (IFNK), which is capable of positively signaling through the type I interferon receptor, promoting the upregulation of other types of interferons that become essential in the development of innate antiviral immunity and contribute to the formation of an inflammatory tumor microenvironment [78]. Moreover, for that, E5 is capable of suppressing the IFN I pathway by interacting with and binding to Mitochondrial Antiviral-Signaling Protein (MAVS) and Stimulator of IFN Genes (STING), with evidence of direct suppression of E5 on the activity of STING agonists in two mouse tumor challenge models [79]. In the aforementioned study, an immunopeptidome analysis demonstrated that E5 reduces the expression and function of the immunoproteasome, which is associated with improved overall survival in patients.

Cyclooxygenase-2 (COX-2) is also widely reported as an enzyme capable of modulating the tumor microenvironment through its activation by the E5 oncoprotein, being responsible for producing type 2 prostaglandins (PGE2) that can bind to their specific receptors, producing a highly inflammatory tumor microenvironment [80]. Furthermore, HPV E5 is reported in the development of metastases through Met signaling, which contributes to the increased motility of cancer cells in the tumor microenvironment [81]. Therefore, studying the mechanisms involving the alteration of the tumor microenvironment by HPV can be extremely valuable in understanding the role of the E5 oncoprotein in modulating the tumor microenvironment.

### 4.3. Modulation of Apoptosis

Considering the HPV oncoproteins, E6 is more directly associated with cell survival, E7 with proliferation, while E5 supports these oncoproteins, aiding in both mechanisms as well as in apoptosis modulation and viral immune evasion [10]. Studies suggest that E5 may impair cell-to-cell communication and enhance the immortalization effects of E6 and E7 in keratinocytes [82,83].

The E5 oncoprotein can alter apoptotic pathways by inducing the degradation of the pro-apoptotic protein BAK via ubiquitination and by impairing the signaling of FAS ligand and TNF-related apoptosis-inducing ligand (TRAIL), downregulating FAS receptor expression and inhibiting the recruitment of associated proteins to form the death-inducing signaling complex (DISC) [84]. With the inhibition of DISC formation mediated by altered TRAIL signaling, the cleavage of procaspase-8 and -3, as well as PARP, is suppressed [85]. Additionally, a study demonstrated that HPV16 E5 expression in human keratinocytes promotes resistance to apoptosis induced by ultraviolet B (UV-B) radiation [86].

EGFR activation and PI3K-AKT and ERK1/2 MAPK signaling pathways mediated by E5 may also be suggested as mechanisms through which this oncoprotein modulates apoptosis, regulating these key components essential for cell survival [86]. A gene expression analysis demonstrated that E5 can repress the expression of endoplasmic reticulum components related to oxidative stress response pathways and enhance PI3K-AKT signaling pathway expression [87]. Moreover, through EGFR/NF-κB signaling, E5 promotes increased COX-2 expression, leading to substantial prostaglandin production and subsequent apoptosis inhibition [88,89]. Based on these studies, it is evident that E5 can modulate apoptosis through various mechanisms. In HNC, therefore, the mechanisms related to the activity of this oncoprotein in apoptosis modulation need to be evaluated.

### 4.4. Synergy with E6 and E7

It has long been demonstrated that the E5 oncoprotein exhibits transformative activity and can act in synergy with E6 and E7. To assess the activation of cell proliferation mediated by the cooperation between E5 and E7, primary rodent cells were transfected with E5 and E7, resulting in a potent mitogenic response that was enhanced in the presence of epidermal growth factor (EGF). This finding suggests a synergistic action between E5 and E7 in promoting cell proliferation [90]. More recent studies have revealed that HPV-18 E5 can also induce cell proliferation and invasion by moderately increasing oxidative stress through elevated levels of intracellular reactive oxygen species (ROS) in cells expressing E6 and E7 [91]. However, research exploring this cooperation specifically in head and neck cancers remains limited. A study by Ren et al. (2020) [55], using in vitro and in vivo models of head and neck cancers, demonstrated that E5 can cooperate with other early viral proteins, such as E2 and E4, during the initial phase of infection when the viral genome is in its episomal form. In this study, the combined activity of E2, E4, and E5 exhibited transformative potential, activating cell proliferation in a p53-dependent manner, suggesting an alternative pathway for carcinogenesis in HPV-associated head and neck cancers.

## 5. E5 as a Therapeutic Target in Head and Neck Cancer Treatment

HPV infection has emerged as one of the primary etiological factors for HNC [92]. Despite the widespread support for prophylactic vaccines, which primarily provide protection against new infections, and screening programs for cervical cancer, HPV remains a major concern, particularly in developing countries. For this reason, therapeutic vaccination strategies aiming to activate DC and NK cells and stimulate the proliferation of cytotoxic CD8+ T cells targeting HPV oncoproteins and Th1 CD4+ T cells are emerging as promising approaches for the treatment of HPV-positive cancers [93,94].

The E5 protein has been evaluated for its use as a therapeutic target or potential treatment for HPV-related cancers in animal models [95,96,97,98,99,100]. A significant development in this area has been the use of a monoclonal antibody conjugate designed to target dendritic cells (DCs). In this approach, the E5 protein was linked to an antibody specific to DEC-205 (anti-DEC-205:16E5). This method was shown to enhance the ability of targeted DCs to capture and present E5 HPV16 antigens, leading to a strong protective immune response and offering a promising new direction in immunotherapy for HPV-associated cancers [95].

The use of E5 as a therapeutic target or as therapy has garnered interest, especially in cases where precancerous lesions and tumors are diagnosed in early stages, given its higher expression during this period [99,101,102]. However, E5 expression is considered to be lost or decreased following HPV integration into the cellular genome, though it might be more accurate to note that there are significant variations in E5 detections in some of those reports [103,104,105]. Furthermore, recent studies have demonstrated the role of E5 in maintaining the viral genome in an episomal state, possibly acting as an additional mechanism of escape from the innate immune response [55,78].

The downregulation of MHC I expression caused by E5 HPV is especially relevant in the context of anti-PD-1 therapies. In the PD-1/PD-L1 pathway, the interaction of PD-1 receptors on B and T cells and the PD-L1 ligand on tumor cells is known to cause inhibition of T cell-mediated antitumor immunity [106,107]. Studies have found that HPV-specific CD8+ T cells express PD-1, suggesting that this pathway contributes to inhibitory regulation in HPV-positive tumors. The presence of these cells in tumors has been found in varying amounts, ranging from 0.1% to 10% of tumor-infiltrating CD8+ T lymphocytes [11].

Clinical trials evaluating the efficacy of inhibitors of the PD-1/PD-L1 pathway have shown promising but mixed results in patients with HPV-positive HNSCC, where some did not respond well to these treatments [11,108,109]. While these inhibitors have the potential to reactivate HPV-specific T cells and improve treatment outcomes, their effectiveness can be limited [11].

Moreover, the presence of the HPV E5 oncoprotein can induce significant resistance to anti-PD-1 immunotherapy by reducing MHC expression and interfering with antigen presentation [11]. This finding is further supported by observations in HPV+ HNSCC tumors, where high E5 expression has been associated with poorer survival and pharmacological inhibition. In contrast, preclinical models have reported a tumor response characterized by increased MHC expression and T-cell activation [29,110]. In this context, exploring the mechanisms employed by E5 could assist in the development of new targeted therapies for HPV-positive HNC.

Rimantadine, a licensed influenza A drug [111], has been shown to act as an E5 inhibitor [112,113] and can be used to overcome E5-mediated anti-PD-L1 resistance. Studies in mice support the use of rimantadine in HNC, as results demonstrate the restoration of MHC class I expression, which is associated with increased animal survival [114]. Following this concept, Miyauchi and collaborators demonstrated the antitumoral activity of rimantadine in tumor models expressing E5, with marked MHC upregulation and reduced tumor growth even without the administration of radiation and anti-PD-L1 therapy [29].

E5-mediated HLA retention may lead to low viral antigen expression in tumor cells and immune cell inactivation, representing a limiting factor for the clinical use of immunotherapies in HPV-positive patients [115,116]. However, another hypothesis suggests that HLA downregulation may be compensated by the elimination of HPV-positive cancer cells by natural killer cells [117]. Therefore, further investigation is required to elucidate the underlying mechanisms and the potential application of E5-targeted therapies.

Recently, a study demonstrated that the combination of a short interfering RNA targeting E5 (siRNA-E5) with Oxaliplatin or Ifosfamide enhances chemotherapy efficacy in HPV16-positive cervical cancer cells. In this study, a reduction in the number of colonies was observed, along with decreased expression of genes related to stemness and cell migration in the groups receiving the combined treatment. The authors suggest that using E5-siRNA may allow for lower chemotherapy doses, reducing toxicity while maintaining therapeutic efficacy [118].

These findings highlight the importance of the HPV16 E5 protein in treatment resistance and suggest that additional research into other viroporin inhibitors in E5 tumor models, as well as the combination of rimantadine with other immunological and radiation therapies, could be an effective strategy for improving outcomes for head and neck cancer patients. Understanding how HPV mediates immune suppression is critical for improving the efficacy of immunotherapies and achieving better clinical outcomes in HPV-related cancers.

## 6. Conclusions

HPV infection significantly increases the development of head and neck cancer, particularly in the oropharyngeal subset. Patients with this type of cancer who are HPV-positive generally have a better prognosis; however, chemotherapy resistance in these cases is commonly reported and has been associated with the HPV oncoprotein E5. Studies suggest that HPV E5 promotes immune evasion, which may be related to increased resistance to anti-PD-L1 therapy, currently one of the main treatment methods for HNC. HPV E5 facilitates immune evasion through its interaction with the major histocompatibility complex via the STING signaling pathway and modulation of the tumor microenvironment. In recent years, there has been growing interest in therapies targeting E5, and in vitro studies suggest that E5-targeted therapy may enhance the response to different drugs, leading to fewer side effects for patients undergoing treatment. In this context, studies aimed at elucidating the role of E5 in HNC are essential to support the development of promising new therapies for HPV-positive HNC treatment.

## Figures and Tables

**Figure 1 viruses-17-00512-f001:**
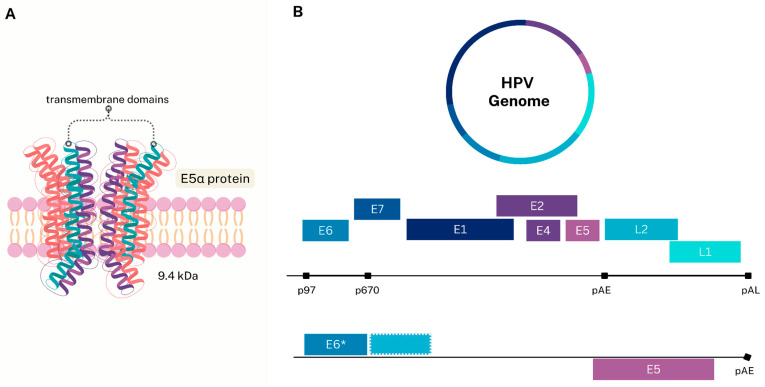
Structure and genome of HPV16 E5 In illustration (**A**), the three hydrophobic transmembrane domains of the HPV16 E5α protein are depicted illustratively. In the adjacent diagram (**B**), the complete HPV16 genome is shown, along with a specific splicing event containing the E5 gene and the truncated initial region of E6 (E6*).

**Figure 2 viruses-17-00512-f002:**
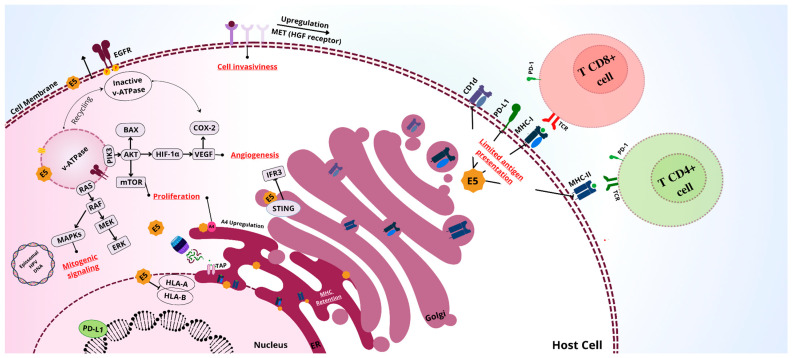
Impact of HPV E5 Oncoprotein on Immune Evasion and Modulation of Cellular Pathways Critical to Oncogenesis. This figure illustrates the multifaceted impact of the HPV E5 oncoprotein on host cells. E5 contributes to immune evasion by limiting antigen presentation through MHC I retention on the host cell, increasing PD-L1 expression, and inhibiting the STING pathway. Furthermore, E5 promotes cell proliferation by activating the PI3K/AKT/mTOR pathway and facilitates angiogenesis by upregulating VEGF expression via HIF-1α. E5 also interferes with mitogenic signaling through activation of the MAPK pathway, promoting cell invasiveness by increasing MET expression.

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
