# Peer review of "E5 Oncoprotein: A Key Player in Human Papillomavirus-Positive Head and Neck Cancer Pathogenesis and Therapy Resistance"

_viruses, 2025, doi:10.3390/v17040512_

Round 1

Reviewer 1 Report

Comments and Suggestions for Authors

In the manuscript by Pereira Santos et al. the authors review the E5 oncoprotein roles in HNSCC. The manuscript is easy to understand and mostly well written. Some issues listed below could be improved

P1 abstract. The authors generalize HNC and HPV positive HNC too much. HPV is a favorable prognostic factor for the oropharyngeal subset. On the other hand, HPV is not a clear cut factor for oral cancer. Thus, some over generalization parts might be revised to be more precise.

P1 39  ref 5 is maybe not the best (not primary) reference for this claim. This problem of not citing primary references is found throughout the manuscript where often the citation is of a review and not the primary study for some claim. For example, the link between E7 and pRb was not first shown in 2023 (P3 L94).

P2 L81 in text referencing of the figure panels is not appropriate. Fig 1b is referenced before Fig 1A  (P3 L 107). The order of Fig 1 panels and legend should be readjusted accordingly

P3 L 105 “The E5 oncoprotein is not present in all HPV types; its gene is restricted to Alphapapillomavirus species,…” is misleading the readers to think only alpha group has EP while in fact some non human PV species especially BPV also contain E5 (Delta, Xi). https://doi.org/10.1016/j.virol.2013.05.012. Ideally this should be mentioned so that readers are aware of this non exclusivity.

P3 L107 and L111 Figure 1 A doesn’t illustrate the point the authors raise at these 2 points in the manuscript

P3 L111 the reference “32. da Silva Ramalho, J.A.; Palma, L.F.; Ramalho, K.M.; Tedesco, T.K.; Morimoto, S. Effect of Botulinum Toxin A on Pain, Bite Force, and Satisfaction of Patients with Bruxism: A Randomized Single-Blind Clinical Trial Comparing Two Protocols. Saudi Dent. J. 2023, 35, 53–60, doi:10.1016/j.sdentj.2022.12.008” seems out of place in the context of “two variants of the E5 protein …32]”

P4 L115 the reference “34 McBride, A.A. The Papillomavirus E2 Proteins. Virology 2013” is misplaced

P4 L 158 the seminal staging publication is missing from the review

https://pubmed.ncbi.nlm.nih.gov/28094848/

P5 L178 the seminal mutational landscape publication is missing from the review

https://www.nature.com/articles/nature14129

P5 L184-188 the section about B cells seems completely out of place at this point

P6 L215 the reference 61 is dealing with cervical cancer which should be acknowledged (for this and other such references)

P7 L 237-239 the abbreviations GA and ER are used only a few times and only at this point in the manuscript and could be removed

P8 L317 technical problem with references “[96–100,100,101]” should be “[96-101]”

Author Response

REPLY TO REFEREES

Recife, March 26, 2025.

Dear Reviewer,

I am re-submitting the review article entitled “E5 Oncoprotein: A Key Player in HPV-Positive Head and Neck Cancer Pathogenesis and Therapy Resistance” by Pereira Santos and São Marcos et al., for publication in Viruses.

All modifications are highlighted in the manuscript.

All the authors confirm that they saw and agreed to the submitted paper. The authors have been recognized as contributors and have agreed to their inclusion. The material is original, and it has been neither published elsewhere nor submitted for publication simultaneously. None of the authors has any potential financial conflict of interest related to this manuscript.

Responses

1) P1 abstract. The authors generalize HNC and HPV positive HNC too much. HPV is a favorable prognostic factor for the oropharyngeal subset. On the other hand, HPV is not a clear cut factor for oral cancer. Thus, some over generalization parts might be revised to be more precise.

Answer: We have corrected the generalization throughout the text, highlighting oropharyngeal cancer as more strongly associated with HPV.

2) P1 39  ref 5 is maybe not the best (not primary) reference for this claim. This problem of not citing primary references is found throughout the manuscript where often the citation is of a review and not the primary study for some claim. For example, the link between E7 and pRb was not first shown in 2023 (P3 L94).

Answer: We have addressed this issue throughout the manuscript.

3) P2 L81 in text referencing of the figure panels is not appropriate. Fig 1b is referenced before Fig 1A (P3 L 107). The order of Fig 1 panels and legend should be readjusted accordingly.

Answer: We have corrected the figure citations and added them in the appropriate places.

4) P3 L 105 “The E5 oncoprotein is not present in all HPV types; its gene is restricted to Alphapapillomavirus species,…” is misleading the readers to think only alpha group has EP while in fact some non human PV species especially BPV also contain E5 (Delta, Xi). https://doi.org/10.1016/j.virol.2013.05.012. Ideally this should be mentioned so that readers are aware of this non exclusivity.

Answer: We have corrected the information and added the corresponding references.

5) P3 L107 and L111 Figure 1 A doesn’t illustrate the point the authors raise at these 2 points in the manuscript

Answer: We have added the figure citation in the correct place and removed the previous one.

6) P3 L111 the reference “32. da Silva Ramalho, J.A.; Palma, L.F.; Ramalho, K.M.; Tedesco, T.K.; Morimoto, S. Effect of Botulinum Toxin A on Pain, Bite Force, and Satisfaction of Patients with Bruxism: A Randomized Single-Blind Clinical Trial Comparing Two Protocols. Saudi Dent. J. 2023, 35, 53–60, doi:10.1016/j.sdentj.2022.12.008” seems out of place in the context of “two variants of the E5 protein …32]”

Answer: We have removed the erroneously added reference.

7) P4 L115 the reference “34 McBride, A.A. The Papillomavirus E2 Proteins. Virology 2013” is misplaced

Answer: We have corrected the references for this information.

8) P4 L158 the seminal staging publication is missing from the review https://pubmed.ncbi.nlm.nih.gov/28094848/

Answer: We have properly added the reference.

9) P5 L178 the seminal mutational landscape publication is missing from the review  https://www.nature.com/articles/nature14129

Answer: We have properly added the reference.

10) P5 L184-188 the section about B cells seems completely out of place at this point

Answer: We have removed the section on this topic.

11) P6 L215 the reference 61 is dealing with cervical cancer which should be acknowledged (for this and other such references)

Answer: We have added this important information.

12) P7 L 237-239 the abbreviations GA and ER are used only a few times and only at this point in the manuscript and could be removed

Answer: We have removed the acronyms and corrected the text to include the full names associated with these acronyms.

13) P8 L317 technical problem with references “[96–100,100,101]” should be “[96-101]”

Answer: We have corrected the citations of these references.

Please do not hesitate to contact me if further information is needed.

We appreciate your insightful comments, which have helped improve the clarity and precision of our work.

Sincerely,

ANTONIO CARLOS DE FREITAS, PH.D

Associate Professor

Head of Laboratory of Molecular Studies and Experimental Therapy (LEMTE)

Department of Genetics

Federal University of Pernambuco

Recife, Pernambuco -Brazil

Reviewer 2 Report

Comments and Suggestions for Authors

This manuscript is a solid attempt to explore the role of the HPV E5 oncoprotein in the pathogenesis and therapy resistance of HPV-positive head and neck cancer. I would suggest several improvements before it can be accepted for publication in the journal Viruses. There is some redundancy in the manuscript that can be considered for a removal, as it overextends the text - for example, the section on HPV integration covers material more suited to a general HPV review, not one specifically emphasizing E5.  Similarly, the repetitive mention of E6 and E7 mechanisms detracts from the review's stated objective.

What is missing is clear new perspective or identify key gaps in current knowledge, which diminishes the potential impact and usefulness of this review. For example, not enough is given in the text for E5's role in therapy resistance and immune evasion. The manuscript outlines known mechanisms, such as E5's interference with MHC class I molecules and PD-L1 modulation, but stops short of analyzing conflicting data or heterogeneity in E5 expression across patients. There are also more challenges in therapeutically targeting E5, which should be addressed. There is no discussion of why (despite extensive E5 research) therapeutic interventions remain limited, nor an assessment of whether targeting E5 is realistically feasible in clinical settings. Emerging data on differential E5 expression levels, particularly in HNC versus cervical cancer, should be critically addressed.

The section discussing potential therapeutic interventions also largely reiterates results from a few selected studies without a broader exploration of possible small-molecule inhibitors, immune strategies or combinatory treatments. There is no discussion of the pharmacological challenges of targeting a small, membrane-bound viral protein like E5, nor is there any mention of ongoing clinical trials or translational barriers. The brief reference to rimantadine, for example, is provided superficially without critically evaluating its potential limitations or clinical applicability.

Additionally, the paper lacks a proper discussion of limitations in the existing literature. I would argue there is a lot of heterogeneity in HPV-positive HNC cohorts, publication bias, variation in viral genome integration status, differences in immune microenvironment profiles, etc. etc. All of this should be addressed. Finally, the conclusion section should provide specific research directions for this field.

Author Response

Responses

1) There is some redundancy in the manuscript that can be considered for a removal, as it overextends the text - for example, the section on HPV integration covers material more suited to a general HPV review, not one specifically emphasizing E5.  Similarly, the repetitive mention of E6 and E7 mechanisms detracts from the review's stated objective.

Answer: We have removed the detailed information on E6 and E7 and expanded the literature on E5. The added information is highlighted in the text.

2) What is missing is clear new perspective or identify key gaps in current knowledge, which diminishes the potential impact and usefulness of this review. For example, not enough is given in the text for E5's role in therapy resistance and immune evasion. The manuscript outlines known mechanisms, such as E5's interference with MHC class I molecules and PD-L1 modulation, but stops short of analyzing conflicting data or heterogeneity in E5 expression across patients. There are also more challenges in therapeutically targeting E5, which should be addressed.

Answer: We have expanded this discussion in the immune evasion section and also explained the studies related to the differential expression of E5. The added information is highlighted in the text.

3) There is no discussion of why (despite extensive E5 research) therapeutic interventions remain limited, nor an assessment of whether targeting E5 is realistically feasible in clinical settings. Emerging data on differential E5 expression levels, particularly in HNC versus cervical cancer, should be critically addressed.

Answer: We have included this information in the section on E5 activity in head and neck cancer and in the section on therapeutic activities of E5. The added information is highlighted in the text.

4) The section discussing potential therapeutic interventions also largely reiterates results from a few selected studies without a broader exploration of possible small-molecule inhibitors, immune strategies or combinatory treatments.

Answer: We have included this information in the section on E5 activity in head and neck cancer and in the section on therapeutic activities of E5. The added information is highlighted in the text.

5) There is no discussion of the pharmacological challenges of targeting a small, membrane-bound viral protein like E5, nor is there any mention of ongoing clinical trials or translational barriers. The brief reference to rimantadine, for example, is provided superficially without critically evaluating its potential limitations or clinical applicability.

Answer: We have included this information in the section on therapeutic activities of E5. The added information is highlighted in the text.

6) Additionally, the paper lacks a proper discussion of limitations in the existing literature. I would argue there is a lot of heterogeneity in HPV-positive HNC cohorts, publication bias, variation in viral genome integration status, differences in immune microenvironment profiles, etc. etc. All of this should be addressed.

Answer: We have included this information in the section on E5 activity in head and neck cancer. The added information is highlighted in the text.

7) Finally, the conclusion section should provide specific research directions for this field.

Answer: We have revised the conclusion to encompass the indicated aspects.

Please do not hesitate to contact me if further information is needed.

We appreciate your insightful comments, which have helped improve the clarity and precision of our work.

Sincerely,

ANTONIO CARLOS DE FREITAS, PH.D

Associate Professor

Head of Laboratory of Molecular Studies and Experimental Therapy (LEMTE)

Department of Genetics

Federal University of Pernambuco

Recife, Pernambuco -Brazil
